# Mesencephalic representations of recent experience influence decision making

John A Thompson[1,2]*, Jamie D Costabile[2], Gidon Felsen[2]

[1]Department of Neurosurgery, University of Colorado School of Medicine, Aurora, United States; [2]Department of Physiology and Biophysics, University of Colorado School of Medicine, Aurora, United States

**Abstract** Decisions are influenced by recent experience, but the neural basis for this phenomenon is not well understood. Here, we address this question in the context of action selection. We focused on activity in the pedunculopontine tegmental nucleus (PPTg), a mesencephalic region that provides input to several nuclei in the action selection network, in well-trained mice selecting actions based on sensory cues and recent trial history. We found that, at the time of action selection, the activity of many PPTg neurons reflected the action on the previous trial and its outcome, and the strength of this activity predicted the upcoming choice. Further, inactivating the PPTg predictably decreased the influence of recent experience on action selection. These findings suggest that PPTg input to downstream motor regions, where it can be integrated with other relevant information, provides a simple mechanism for incorporating recent experience into the computations underlying action selection.

## Introduction

Selecting actions in a dynamic environment should take into account both sensory input and internally-generated estimates of action value. Integrating these sources of information is well-described by a Bayesian framework in which estimates of action value are continually updated by incoming sensory information in order to select the most valuable action (*Körding and Wolpert, 2006*; *Gold and Shadlen, 2007*; *Kim and Basso, 2010*). This updating of action values, based on experiencing the outcomes associated with past actions, is thought to be mediated primarily by striatal circuits (*Lau and Glimcher, 2007*, *2008*; *Histed et al., 2009*; *Tai et al., 2012*; *Kim et al., 2013*). These action values can be maintained in striatal activity and ultimately used to bias activity in downstream motor centers such that the most valuable actions are more likely to be selected (*Hikosaka et al., 2006*, *2014*). This system is capable of flexibly encoding action value estimates in arbitrarily complex and dynamic contexts over a range of time-scales. However, in environments in which only the recent past is relevant to action value, which is often the case in the real world, a simpler complementary mechanism would be to maintain short-term representations of the most recent actions and their outcomes that directly modulate the action selection process.

We studied this possibility by examining behavior and neural activity in well-trained mice performing a task requiring them to select an action based on the dominant component of an odor mixture (*Uchida and Mainen, 2003*). We have previously shown that the superior colliculus (SC) plays a critical role in selecting the action – a leftward or rightward orienting movement – required by this task (*Felsen and Mainen, 2008*, *2012*; *Stubblefield et al., 2013*), consistent with its role in selecting orienting movements in other species (*Glimcher and Sparks, 1992*; *Horwitz and Newsome, 2001*; *Bergeron et al., 2003*; *Krauzlis et al., 2004*; *Song et al., 2011*; *Wolf et al., 2015*). In this study, we asked whether the pedunculopontine tegmental nucleus (PPTg), a mesencephalic sensorimotor hub that provides direct input to the SC (*Graybiel, 1978*; *Beninato and Spencer, 1986*;

*For correspondence: john.a.thompson@ucdenver.edu

**Competing interests:** The authors declare that no competing interests exist.

**eLife digest** The decisions we make are influenced by recent experience, yet it is not known how this experience is represented in the brain. For decisions about when, where and how to move, researchers have hypothesized that recent experience might influence activity in a region of the brainstem – the central trunk of the brain – that is known to be involved in movement.

When deciding when, where and how to move, several areas of the brain are involved in selecting the optimal action. Recent studies suggest that groups of neurons known as locomotor brainstem nuclei may also contribute to making decisions about movements.

Thompson et al. investigated whether a brainstem locomotor area called the pedunculopontine tegmental (PPTg) nucleus in mice might contribute to decision making rather than just conveying the selected response. The mice were trained to recognize particular odors and move to either the left or right to collect a food reward. While the mice were selecting an action, the activity of neurons in the PPTg nucleus reflected the action they had chosen on a previous experience and the outcome of that choice (i.e. whether they received a reward). These representations of past experiences influenced the upcoming decision the mice were about to take.

The findings of Thompson et al. suggest that the PPTg nucleus might play a critical role in the process of selecting the optimal action. Future work will examine what kinds of information about the environment or recent experience have the biggest effect on the activity of this region.

*Stubblefield et al., 2015*), encodes information about recent actions and their outcomes by recording from individual neurons in behaving mice. While numerous regions provide input to the SC (*Sparks and Hartwich-Young, 1989*), many of which may modulate its processing underlying action selection (*Wolf et al., 2015*), the PPTg holds particular interest because it is engaged by sensorimotor tasks across species (*Matsumura et al., 1997*; *Dormont et al., 1998*; *Kobayashi and Isa, 2002*; *Kobayashi et al., 2002*; *Okada and Kobayashi, 2009*; *Norton et al., 2011*; *Thompson and Felsen, 2013*; *Lau et al., 2015*).

We found that actions in this task are influenced by actions and outcomes in the recent past. Further, at the time of action selection and even throughout much of the trial, most PPTg neurons represented the choice (left or right movement), outcome (rewarded or non-rewarded), or both, on the previous trial. Furthermore, we found that these representations influenced action selection: the probability of particular upcoming choices was predictably related to the firing rates of neurons selective for choice on the previous trial, and pharmacological inactivation of the PPTg causally affected behavior, in part by decreasing the influence of recent choices on upcoming choices. Our results suggest a novel mechanism, subserved by the PPTg, for efficiently modulating action selection based on the recent history of actions and their outcomes.

## Results

### Influence of previous trials on behavior

Our overall goal was to address the role played by the PPTg within the interconnected network of brain regions responsible for selecting actions (*Gold and Shadlen, 2007*; *Wolf et al., 2015*). Given its connectivity with the basal ganglia and input to the SC, we hypothesized that PPTg activity may represent recent information that could be relevant to the selection of upcoming choices. To test this hypothesis, we examined first whether behavior in our task was influenced by the choices (left or right) and outcomes (rewarded or non-rewarded) on previous trials, and next whether PPTg activity similarly reflected these previous choices and outcomes.

Towards this end, we trained seven mice on a well-established odor-guided spatial choice task (*Uchida and Mainen, 2003*; *Thompson and Felsen, 2013*). In each trial of the task, the mouse samples a binary odor mixture at a central port, waits for a go signal, and reports which odor is dominant by moving to the left or right port for a water reward (*Figure 1A*; see Materials and methods). We have previously used this task to show that the activity of a population of PPTg neurons tends to be higher preceding contralateral than ipsilateral orienting movements (*Thompson and Felsen,*

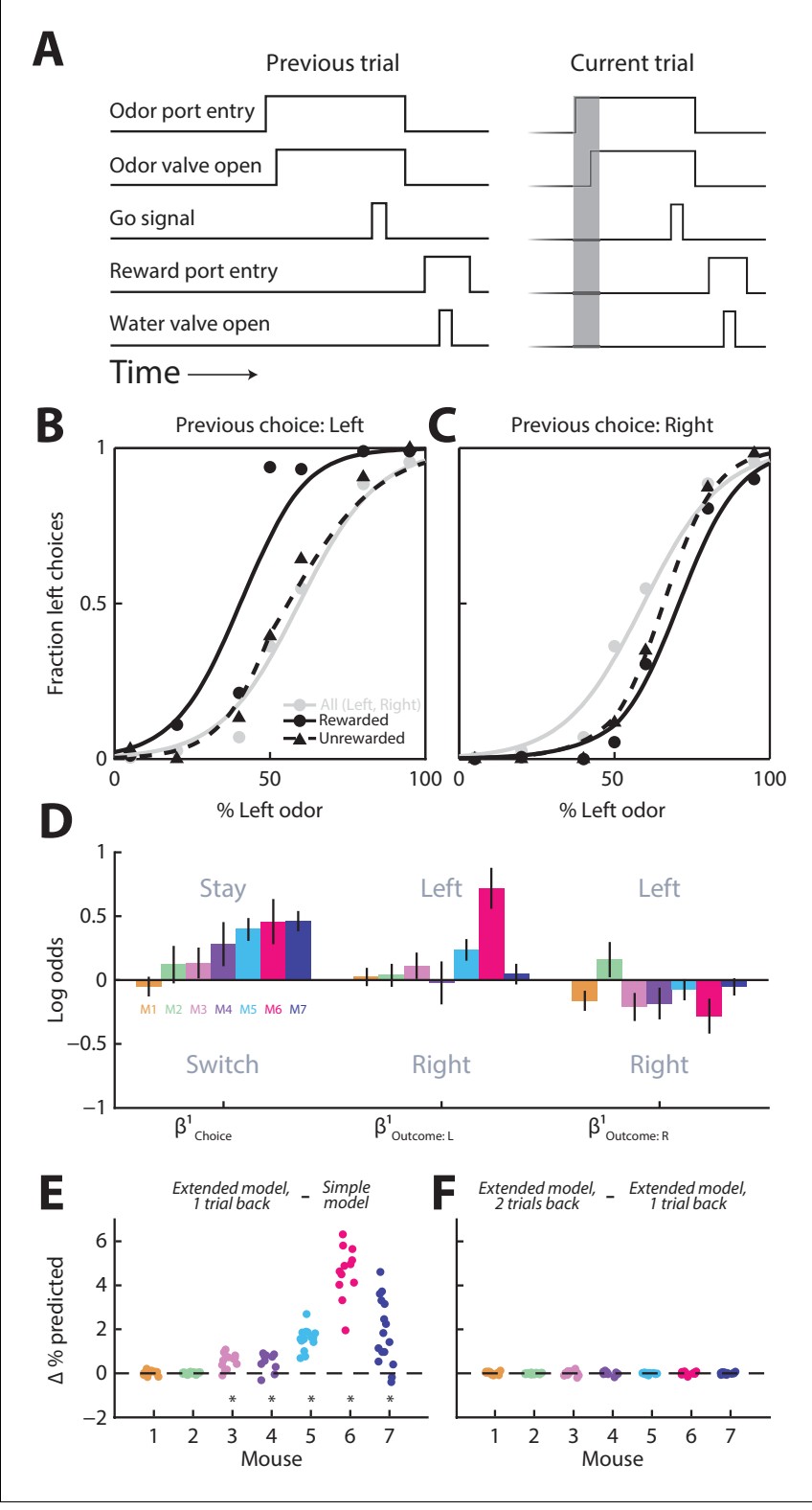

**Figure 1.** Behavior is influenced by previous trials. (**A**) Timing of behavioral events for two consecutive trials ('previous' and 'current') of the spatial choice task. Gray box shows pre-stimulus epoch, the primary focus of our neurophysiological analyses. (**B** and **C**) Behavioral performance conditional on whether the choice on the previous trial was left (**B**) or right (**C**), and rewarded or unrewarded, for 1 example mouse (12 sessions). Gray circles and lines include all trials and are identical in **B** and **C**, shown for comparison with the conditional data (black). Lines

*Figure 1 continued on next page*

*Figure 1 continued*

show best-fit logistic functions using the *Simple model*. (D) Influence of previous choice and outcome on choice behavior for 7 mice (M1-M7), estimated with the *Extended model, 1 trial back*. Error bars, 95% confidence intervals. (E and F) Improvement in predicting choices realized by including in the behavioral model choices and outcomes 1 trial back (E) and 2 trials back (F) for all seven mice (corresponding to M1-M7 in D). Each symbol represents 1 session. For each session, simulated choices on a test set of trials (50% of trials in the session) were separately generated by three models (*Simple model; Extended model, 1 trial back; and Extended model, 2 trial back*) in which all regression coefficients were estimated from the remaining trials (in all sessions) for that mouse. The accuracy of each model's prediction of choices in the test set was calculated as the percentage of trials in which the predicted choice matched the actual choice. This process was repeated, with a new test set, 50 times/session. Each symbol shows the average, across repeats for each session, of the improvement in accuracy of the *Extended model, 1 trial back* over the *Simple model* (E), and of the *Extended model, 2 trials back* over the *Extended model, 1 trial back* (F). *p<0.0001, one-tailed paired Student's *t* test across sessions per mouse.

*2013*). Although optimal performance on this task requires that choices be based only on the current stimulus, previous studies – in mice, primates and humans – have shown that behavior in such tasks is, nevertheless, often influenced by previous trials (*Lau and Glimcher, 2005*; *Gold et al., 2008*; *Busse et al., 2011*; *Akaishi et al., 2014*). We therefore examined how, in our task, upcoming choices depend on the choices and outcomes of previous trials by first examining psychometric functions conditional on previous trial history. As illustrated in the example data from one mouse in *Figure 1B,C*, the mouse tended to choose the left port more often when it had chosen the left port on the previous trial and had been rewarded (*Figure 1B*, compare solid black line to gray line), and tended to choose the right port more often when it had chosen the right port on the previous trial whether or not it had been rewarded (*Figure 1C*, compare solid and dashed black lines to gray line). To quantify the influence of previous trials on choices, separately for each mouse (12 sessions per mouse; 451 ± 115 (mean ± STD) trials per session), we employed a logistic regression model with terms for previous choice ($\beta^1_{Choice}$) and outcome ($\beta^1_{Outcome: L}$ and $\beta^1_{Outcome: R}$) (*Extended model, 1 trial back*; see Materials and methods). The sign of $\beta^1_{Choice}$ reflects the tendency to stay with the same (positive) or switch to the opposite (negative) choice as on the previous trial, while the sign of $\beta^1_{Outcome: L}$ and $\beta^1_{Outcome: R}$ reflects the tendency to choose left (positive) or right (negative) when the previous choice was rewarded at the side indicated in the subscript. Consistent with the example data shown in *Figure 1B,C*, and with some previous work (*Lau and Glimcher, 2005*), we found that, while there was some variability across mice, they tended to choose the same side as they had on the previous trial (reflected in positive values of $\beta^1_{Choice}$) [although in different tasks, a tendency to choose the opposite side has been observed (*Lau and Glimcher, 2005*; *Kim et al., 2007*; *Sul et al., 2011*)], and that this tendency was enhanced when the mouse was rewarded on the previous trial (reflected in positive values of $\beta^1_{Outcome: L}$ and negative values of $\beta^1_{Outcome: R}$; *Figure 1D*). In order to determine whether including these terms improved our behavioral model, we compared the accuracy of the *Extended model, 1 trial back* to the accuracy of the *Simple model* at predicting choices (Materials and methods). We found that including the choice and outcome on the previous trial improved the predictive accuracy of the model for five of seven mice (*Figure 1E*, p<0.0001, one-tailed paired Student's *t* test separately by mouse). However, including the choice and outcome from two trials back did not further improve the predictive accuracy of the model in any mice (*Figure 1F*, p>0.16, one-tailed paired Student's *t* test).

In principle, the influence of the previous choice on behavior (*Figure 1E*) could be mediated in part by body position. For example, if the orientation at which the mouse enters the odor port depends on its previous choice, its subsequent choice behavior may be biased. If this were the case, we reasoned that would likely, although not necessarily, observe that movement duration (from odor port exit to reward port entry) would depend on the previous choice. However, we found that this was not the case (durations for rightward movements preceded by right choices and preceded by left choices, analyzed separately for each mouse, did not differ in 7/7 mice; durations for leftward movements preceded by right choices and preceded by left choices did not differ in 6/7 mice; p>0.05, two-tailed unpaired Students *t*-tests separately by mouse). Therefore, although it is not possible to entirely rule out a role for the position of the body or any of its parts (e.g., whiskers) in

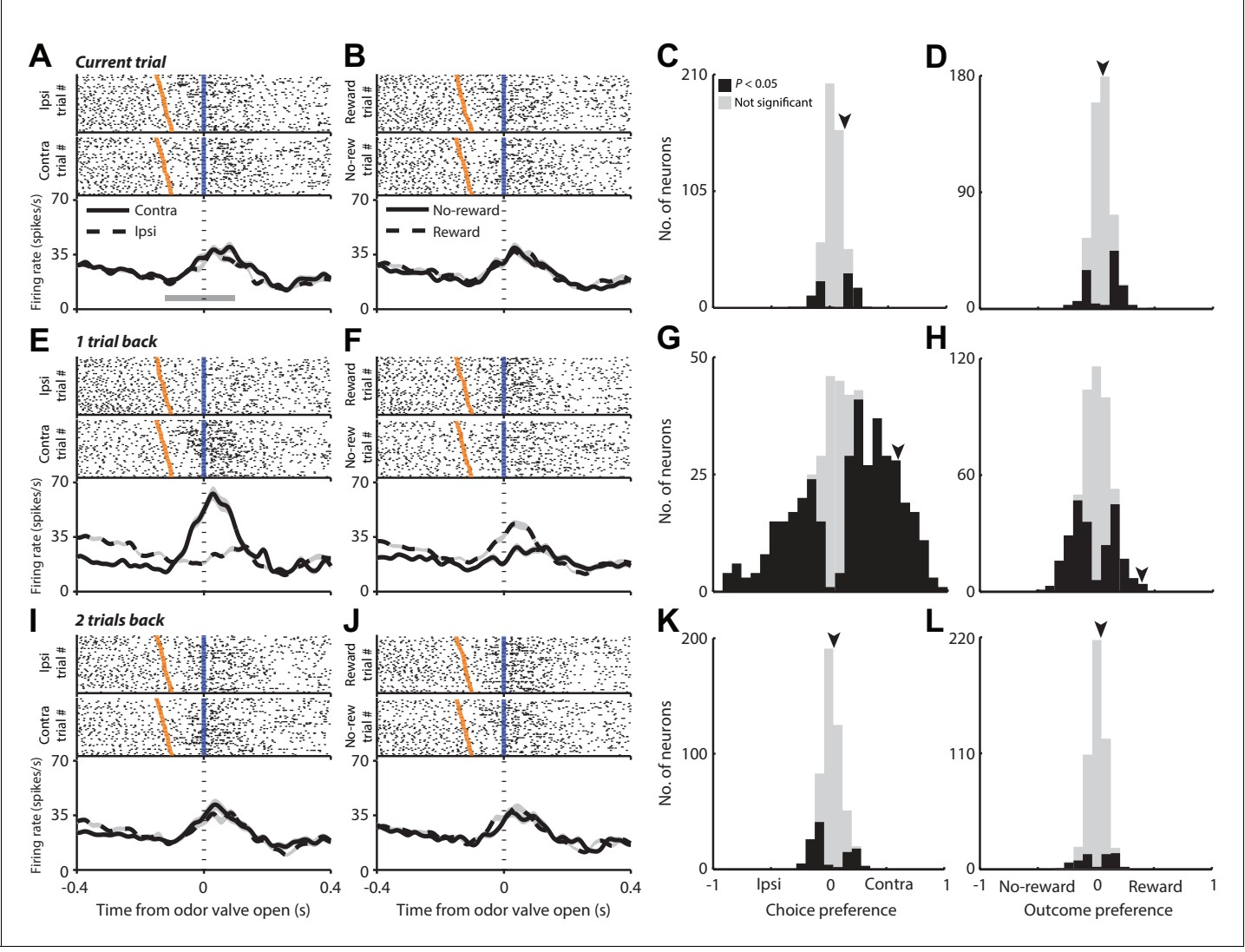

**Figure 2.** PPTg activity during the pre-stimulus epoch is influenced by choices and outcomes on previous trials. (A and B) Rasters (top) and PSTHs (bottom) for one example neuron grouped by choice (A) and outcome (B) on the current trial. Fifty pseudorandomly selected ipsilateral and contralateral trials (A) and reward and no-reward trials (B) are shown in the rasters; all trials are included in the PSTHs. Activity is aligned to the time of odor valve open (blue lines) and sorted within each group by the time since odor port entry (orange ticks). PSTHs show average firing rate across all trials in each group, smoothed with a Gaussian filter ($\sigma$ = 15 ms). Shading, ± s.e.m. Gray bar shows mean pre-stimulus epoch. (C and D) Preference during the pre-stimulus epoch for choice (C) and outcome (D) on the current trial across the population of neurons. Arrowheads indicate preferences for example neuron shown in A and B. (E, F, G and H) As in A, B, C and D with respect to 1 trial back instead of current trial. Data from same example neuron are shown. (I, J, K and L) As in A, B, C and D with respect to two trials back instead of current trial. Data from same example neuron are shown.

contributing to our observed effect, as we did not measure all dimensions of body position, these results reduce the likelihood that choices depend on patterns of body position arising from the previous choice. Together, these results indicate that choices in our task are influenced by what happened on the previous trial – primarily by the choice rather than the outcome – and little influenced by the choice or the outcome from two trials in the past.

## Influence of previous trials on PPTg activity

Having established the relevance of the previous trial for behavior, we next examined PPTg activity recorded in the same seven mice while they performed the task (see Materials and methods). We first asked whether PPTg activity reflects the choice or outcome of the current trial. *Figure 2A* shows

rasters and peristimulus time histograms (PSTHs) for a representative PPTg neuron aligned to the time of odor valve open and grouped by the choice on the current trial, and *Figure 2B* shows the same data grouped by outcome on the current trial. Although the firing rate increased around the time of odor valve open, it does not appear to depend on either the choice (*Figure 2A*) or outcome (*Figure 2B*) on the current trial. To quantify dependence on choice across the population of 506 PPTg neurons recorded, we used an ROC-based preference analysis that uses the firing rates in a particular epoch to assign values ranging from −1 (ipsilateral preference) to 1 (contralateral preference), where the magnitude reflects preference strength (see Materials and methods). We were initially interested in preference during the pre-stimulus epoch – from odor port entry to 100 ms after odor valve open (*Figure 1A*, gray shading; *Figure 2A*, gray bar) – when the mouse is presumably preparing to update its estimates of the value of each option (i.e., to move left or right) with the information provided by the stimulus. We found that few neurons exhibited a preference for choice during this epoch (*Figure 2*, p<0.05, Monte Carlo permutation test). We then used a similar preference analysis to quantify dependence on outcome during this same epoch, where negative values indicate preference for no-reward and positive values indicate preference for reward (see Materials and methods). We found that few neurons exhibited outcome preference (*Figure 2*, p<0.05). It is not surprising that few neurons exhibit choice or outcome preference during this epoch, because the mouse presumably does not commit to a choice and cannot predict the outcome of the trial, before receiving the stimulus.

We next examined whether PPTg activity depended on the choice and outcome on the previous trial, which have been shown to influence behavior (*Figure 1*). *Figure 2E,F* shows data from the same neuron as in *Figure 2A,B* but with trials grouped by the choice (*Figure 2E*) or outcome (*Figure 2F*) on the previous, as opposed to the current, trial. In contrast to the lack of dependence on the choice or outcome on the current trial (*Figure 2A,B*), there appears to be a noticeable difference in firing rate around the time the odor valve opens depending on the choice or outcome on the previous trial (*Figure 2E,F*). We quantified this dependence across the population using the same preference analysis previously described, but with trials grouped according to the choice (*Figure 2G*) or outcome (*Figure 2H*) on the previous trial. Across the population, a sizeable fraction of neurons exhibited a preference for the choice (*Figure 2G*; 26% preferred ipsilateral, 49% preferred contralateral, p<0.05; more preferred contralateral than ipsilateral, p=0.001, $\chi^2$ test) and outcome (*Figure 2H*; 46% were selective, p<0.05) on the previous trial. We found a modest correlation between preference for the choice on the previous (*Figure 2G*) and current (*Figure 2C*) trials (r = 0.25, p=1.42 × 10$^{-8}$).

In principle, the dependence of PPTg activity on previous choice could actually reflect a dependence on the direction of movement from the chosen reward port back to the odor port. While either representation would be useful for conveying information about previous choice, to determine whether PPTg activity reflects choice per se or movement direction, we examined how activity during movement to and from the reward port (which are in opposite directions) depended on which reward port was selected. We found that, rather than representing movement direction, two-thirds of neurons maintained their preference for which reward port was chosen (p=9.12 × 10$^{-6}$, $\chi^2$ test), suggesting that the preference of PPTg neurons for previous choice (*Figure 2G*) cannot be accounted for by a dependence on movement direction.

We then calculated choice and outcome preference with respect to two trials back (*Figure 2I–L*) and found that, as was the case for the current trial, few neurons exhibited a preference for choice (*Figure 2K*) or outcome (*Figure 2L*), consistent with the relatively little influence on behavior of the choice and outcome from two trials in the past (*Figure 1F*). Thus, a larger proportion of neurons exhibited a preference for choice or outcome one trial back than on either the current trial or two trials back (choice, p=2.0 × 10$^{-48}$; outcome, p=3.0 × 10$^{-5}$, Kruskal-Wallis ANOVA; post-hoc comparisons: choice, one trial back vs. current trial, p=5.6 × 10$^{-30}$; one trial back vs. two trials back, p=1.6 × 10$^{-30}$; outcome, one trial back vs. current trial, p=4.0 × 10$^{-4}$; one trial back vs. two trials back, p=0.002), and there were no differences between the fractions of neurons exhibiting a preference for choice or outcome on the current trial and two trials back (choice, two trials back vs. current trial, p=0.92; outcome, two trials back vs. current trial, p = 0.99).

These results suggest that PPTg activity during the pre-stimulus epoch is primarily modulated by the choice and outcome on the previous trial (although our analyses again cannot entirely rule out the possibility that body position also modulates neural activity). Previous work using the same task

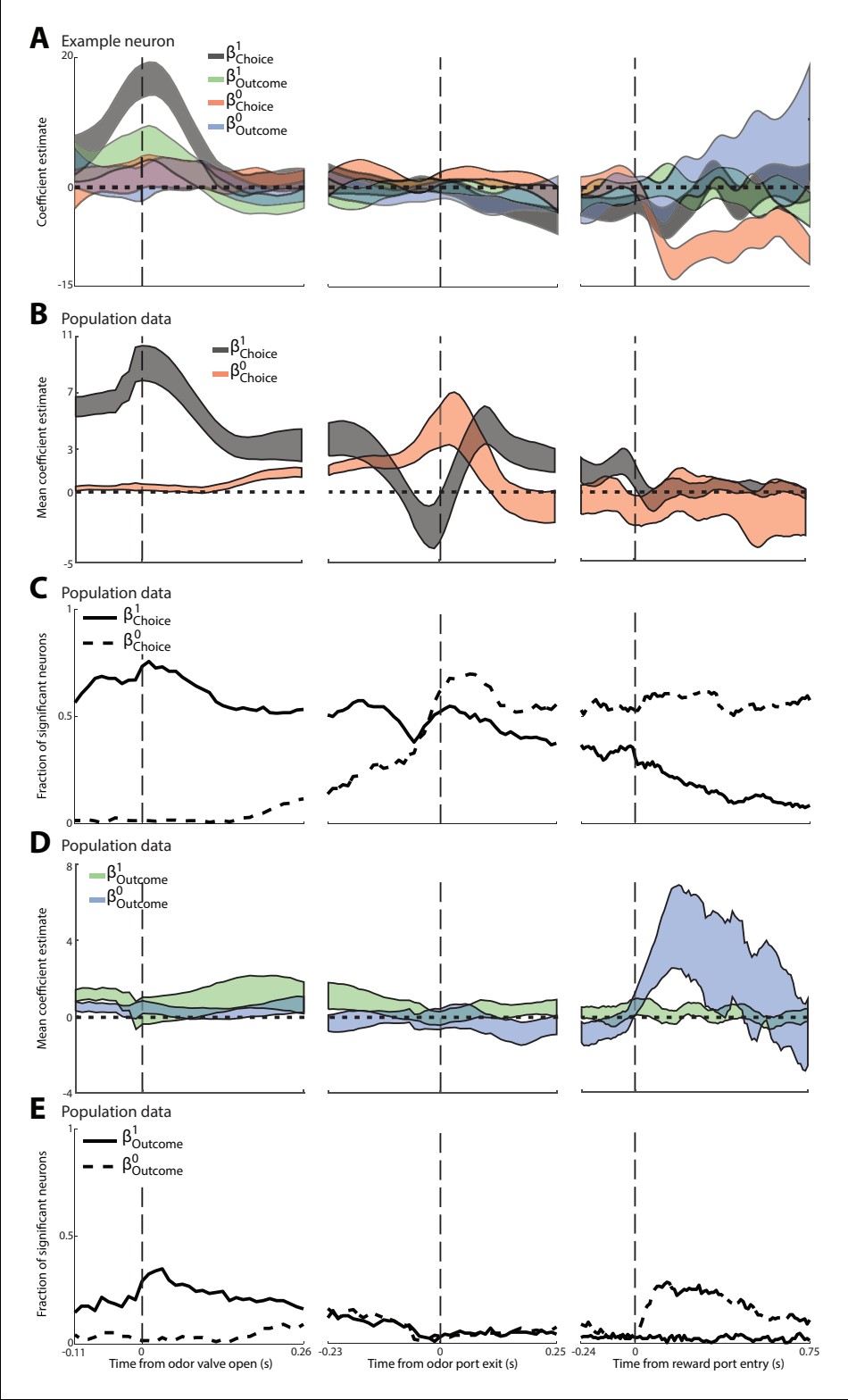

**Figure 3.** Dynamics of the influence of the choice and outcome on the previous and current trial on PPTg activity throughout the trial. (**A**) For the example neuron shown in *Figure 3*, 95% confidence intervals are shown for each regression coefficient calculated in 100 ms bins (shifted by 10 ms), aligned to three trial events. (**B**) Average regression coefficients across all neurons that exhibited a significant preference for choice on the previous trial. Ribbons reflect mean ± SEM $\beta^1_{Choice}$ (black) $\beta^0_{Choice}$ (red) coefficient values as a function of time. (**C**) Fraction of

*Figure 3 continued on next page*

Figure 3 continued

neurons with activity in each 100 ms bin significantly influenced by choice on the previous or current trial, aligned as in **A**. (**D**) As in **B**, with respect to outcome instead of choice. (**E**) As in **C**, with respect to outcome instead of choice.

has shown that PPTg activity in other epochs reflects the choice and outcome on the current trial (*Thompson and Felsen, 2013*), consistent with other studies of PPTg function (*Matsumura et al., 1997*; *Dormont et al., 1998*; *Kobayashi et al., 2002*; *Norton et al., 2011*; *Maclaren et al., 2013*). Together, these results raise the question of how neural activity evolves over the course of the trial from representing previous trial information during the pre-stimulus epoch (*Figure 2G,H*) to current trial information in later epochs. To quantify these temporal dynamics, we performed a linear regression analysis investigating how PPTg activity is influenced, over time, by four factors: previous choice ($\beta^1_{Choice}$), current choice ($\beta^0_{Choice}$), previous outcome ($\beta^1_{Outcome}$) and current outcome ($\beta^0_{Outcome}$) [Materials and methods; we did not include terms for two trials back because this trial had little influence on PPTg activity (*Figure 2K,L*)]. *Figure 3A* shows these $\beta$s as a function of time, for the same example PPTg neuron shown in *Figure 2*. Consistent with the PSTHs shown in *Figure 2*, around the time the mouse enters the odor port (which precedes odor valve open by about 100 ms), the activity of this neuron began to be influenced by previous choice and previous outcome, but not by current choice and current outcome (*Figure 3*, left). The influence of the previous trial declines by the time the mouse exits the odor port (*Figure 3A*, middle), and the influence of the current choice increases upon reward port entry (*Figure 3A*, right), consistent with the fact that many neurons exhibit preference for the current choice during and following entrance to the reward port (*Thompson and Felsen, 2013*).

We performed this regression analysis on each neuron that exhibited a significant preference for choice on the previous trial (corresponding to the black bars in *Figure 2G*; 378/506 total neurons). *Figure 3B and C* show, over the course of the trial, the mean $\beta^1_{Choice}$ and $\beta^0_{Choice}$, as well as the fraction of neurons with activity influenced by the choice on the previous (solid line) and current (dashed line) trial. Initially – even preceding odor delivery – the activity of many neurons reflects the choice on the previous trial, while the activity of virtually no neurons reflects the choice on the current trial (*Figure 3C*, left). During stimulus presentation many neurons begin to reflect the choice on the current trial, and by the time the movement is initiated more neurons reflect the choice on the current than the previous trial (*Figure 3C*, middle). Interestingly, the sign of $\beta^1_{Choice}$ of many neurons briefly inverted immediately preceding movement initiation (*Figure 3B*), consistent with the dip in the fraction of significant neurons at this time (*Figure 3C*). The choice on the current trial retains its influence for the remainder of the trial while the influence of the choice on the previous trial declines (*Figure 3C*, right), although, interestingly, a sizeable fraction of neurons continues to reflect the choice on the previous trial until well-after the mouse has made its choice on the current trial. We also examined, for the same population shown in *Figure 3B,C*, how the fraction of neurons with activity influenced by the outcome on the previous trial (solid line) and current trial (dashed line) changes over the course of the trial (*Figure 3D,E*). Similar to the evolving representation of choice, the activity of more neurons is more strongly influenced by previous trial outcome early in the trial (*Figure 3D,E*, left), and by current trial outcome later in the trial (*Figure 3D,E*, right). In contrast to the long-lasting representation of previous choice (*Figure 3B,C*), virtually none of these neurons represent previous outcome by the time the mouse initiates its movement to the reward port (*Figure 3D,E*, middle; however, note that the population of neurons analyzed was selected based on preference for previous choice, and not previous outcome, during the pre-stimulus epoch). These analyses show how PPTg activity evolves from representing information about the previous trial to representing information about the current trial, which may reflect the transfer of previous-trial information from the PPTg to a downstream region where it can be integrated with sensory evidence in order to form a motor plan.

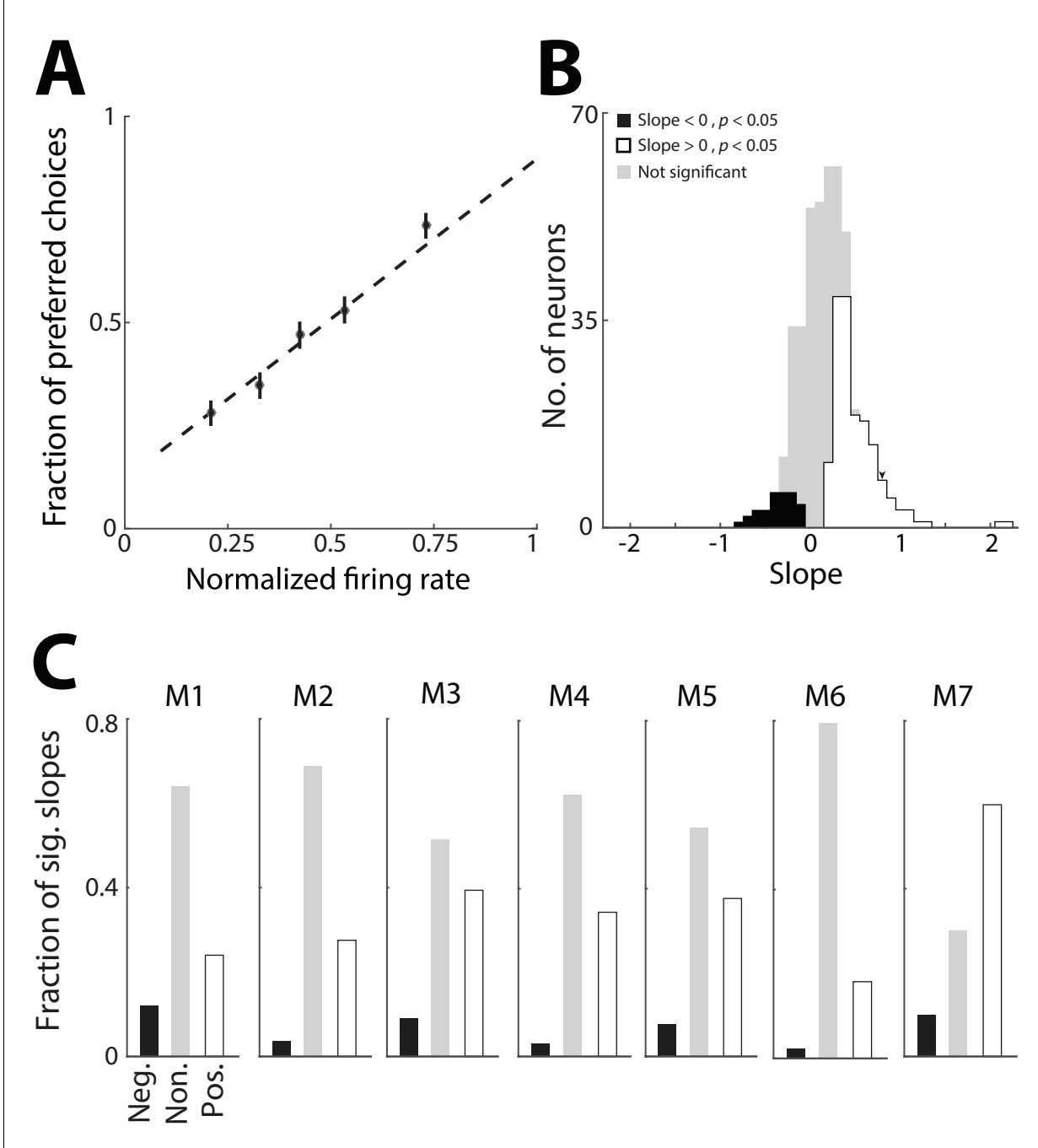

**Figure 4.** PPTg representation of previous choice affects behavior. (**A**) Probability that the mouse chose the reward port corresponding to the previous-trial choice preference of an example neuron, plotted as a function of the firing rate of that neuron during the pre-stimulus epoch. Firing rates were normalized to the maximum for the neuron, separately within each previous choice condition. Only current trials with an ambiguous sensory cue are shown. Dashed line, best linear fit to raw data (antipreferred choice, y = 0; preferred choice, y = 1). Circles represent mean ± s.e.m. fraction preferred choices for binned normalized firing rates and are shown for display only (not used to fit the line). (**B**) Slope of best-fit line calculated as in **A** for all neurons exhibiting a significant preference for choice on the previous trial during the pre-stimulus epoch. Slopes were determined to be significantly different from 0 (p<0.05) with a Monte Carlo permutation test with 1000 repeats. Significance of slopes was identical when the data were fit to a logistic function rather than a line as shown in **A**. Arrowhead indicates slope for example neuron shown in **A**. (**C**) Fraction of significantly positive (white) and negative (black) slopes, and non-significant (gray) slopes (calculated as in **B**), separately for each mouse, numbered as in *Figure 2*.

## Influence of PPTg representations of previous trials on behavior

In order to address whether the PPTg activity representing previous trial information is used to guide behavior, we next examined whether this activity influences the choice on the current trial. For simplicity we focused on PPTg activity representing previous choice, and not outcome, because we found that the former representation was more robust (*Figure 2G,H*). If this activity indeed contributes to behavior – e.g., by providing information about the value of each option – we would predict a trial-by-trial correlation between the activity of individual neurons that exhibit a preference for previous choice and the likelihood of the mouse choosing the corresponding reward port, particularly on trials in which the sensory cue is ambiguous. Specifically, on trials in which a given neuron exhibits higher activity during the pre-stimulus epoch, we would expect the mouse to be more likely to choose the reward port corresponding to the preferred choice (ipsilateral or contralateral) of that neuron. Therefore, for each neuron exhibiting a significant preference for choice on the previous trial (corresponding to the black bars in *Figure 2G*; 378/506 total neurons), we examined how choice on ambiguous trials (% Left odor = 40, 50, or 60; *Figure 1B*) depended on firing rate during the pre-stimulus epoch (*Figure 4A*). Specifically, for each trial of the session we calculated the firing rate (normalized to its maximum across trials, separately for each previous choice condition) and classified it by whether the mouse chose the preferred or antipreferred choice of the neuron, quantified as 1 and 0, respectively. We then calculated the slope of the best-fit line through these points, each of which correspond to a trial. For a given neuron, a positive slope indicates that the mouse is more likely to choose the reward port corresponding to the preferred choice of the neuron on trials in which its activity is high during the pre-stimulus epoch, as we had predicted. Across this population of neurons, many more exhibited a significantly positive slope than negative slope (p<0.05, Monte Carlo permutation test; positive slopes: 136/378; negative slopes: 30/378; positive > negative, p=0.001, $\chi^2$ test; *Figure 4B*), indicating a similar relationship between firing rate and current choice as shown in *Figure 4A*. However, since firing rate and current choice both depend on previous choice (*Figuers 1* and *2*), it is possible that this relationship is indirect. To determine whether this was the case, we examined the distribution of slopes separately for each mouse and found that each exhibited more neurons with positive than negative slopes (*Figure 4C*), even though the behavior of some mice did not depend on previous choice (*Figure 1E*). These results suggest that the representation of the choice on the previous trial (*Figure 2*) can directly influence the choice made by the mouse on the current trial.

To test the causality of this relationship, we next asked whether inactivating the PPTg, with muscimol (a GABA$_A$ agonist) would change the degree to which the previous choice influences behavior. We first examined the direct effect on behavior of unilateral PPTg inactivation (see Materials and methods). We found that choices were biased ipsilateral to the inactivated PPTg, as compared to the preceding and following control sessions in which saline was infused to the same PPTg (*Figure 5A*). We quantified this effect across all 30 sets of sessions [saline (pre), muscimol, and saline (post)] from three mice by estimating the influence of muscimol on choice (represented by $\beta_{Muscimol}$; see Materials and methods). Positive values of $\beta_{Muscimol}$ correspond to an ipsilateral influence and negative values correspond to a contralateral influence. We found that unilaterally inactivating the PPTg resulted in a modest ipsilateral influence on choices (*Figure 5B*; p<0.05, two-tailed unpaired Student's *t* test, pooled across 3 animals and 30 sets of sessions (424 ± 105 trials; mean ± STD); $\beta_{Muscimol} > 0$ in 14/30 individual sets of sessions, $\beta_{Muscimol} < 0$ in 6/30 individual sets of sessions, p<0.05, two-tailed one-sample Student's *t* test; black bars), indicating a causal relationship between PPTg activity and contralateral movements in this task. Consistent with the modest but significant effect of unilateral inactivation on choice bias, we found that movement latency and duration – both to and from the reward port – were longer in muscimol than in saline sessions, even when contralateral and ipsilateral choices were pooled (Materials and methods; p=4.55 × 10$^{-20}$, 5.24 × 10$^{-32}$, and 0.0026, respectively, two-tailed unpaired Student's *t* tests).

Finally, we wondered whether the effect on behavior of inactivating PPTg could be due to a decreased dependence of current choice on previous choice. We tested this idea by estimating $\beta^1_{Choice}$ separately for the muscimol and saline sessions and found that $\beta^1_{Choice}$ was indeed lower when PPTg activity was inhibited (*Figure 5C*, p=0.021, one-tailed Mann-Whitney *U* test), perhaps accounting for the modest effect on choices shown in *Figure 5B* [with respect to the other terms in the model, we found that $\beta_{Odor: L}$ and $\beta_{Odor: R}$ did not differ between muscimol and saline sessions

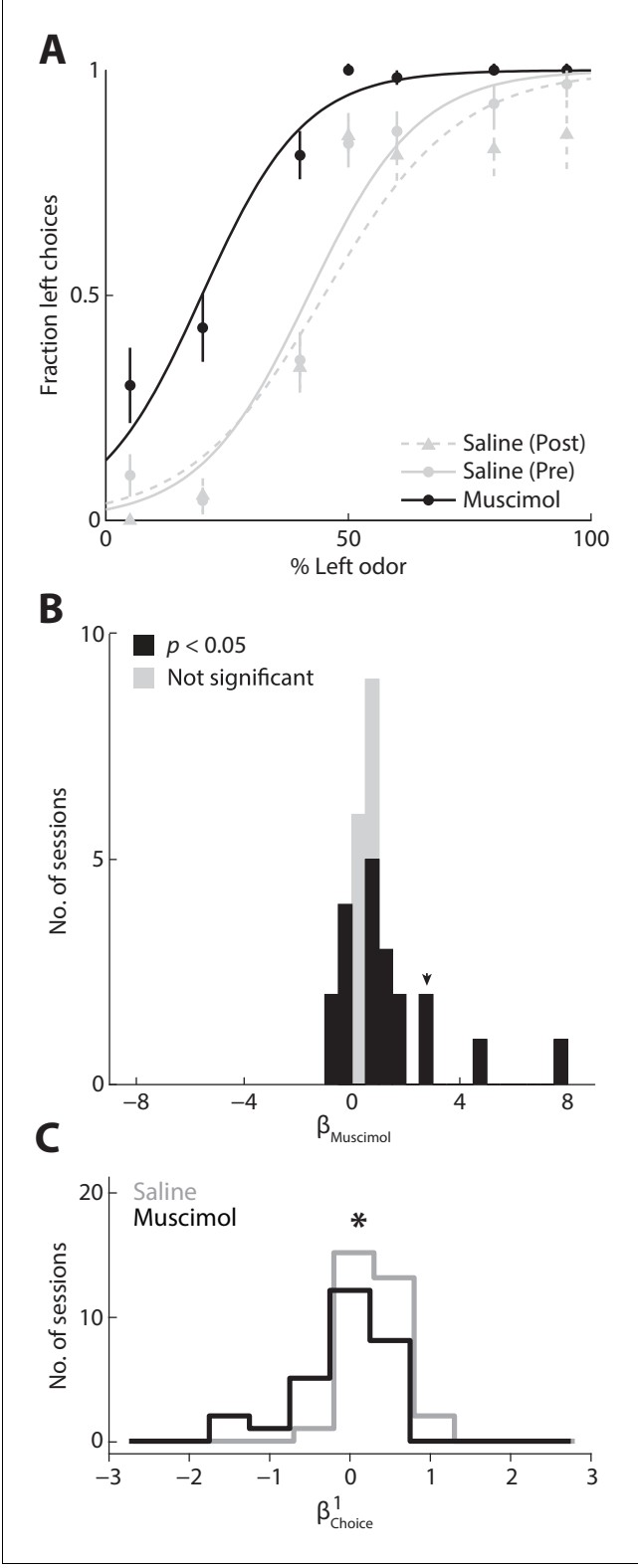

**Figure 5.** Inactivating the PPTg causally affects behavior. (**A**) Behavioral performance during example sessions after which either muscimol (black circles) or 0.9% saline (gray circles) was infused into the left PPTg. Saline was infused during the session before (gray circles, solid gray line) and after (gray triangles, dashed gray line) the muscimol session. One session was performed per day. Lines show best-fit logistic functions using the *Simple model*. Error bars, ± s.d. Behavioral data shown here and in all subsequent figures were collected from well-

*Figure 5 continued*
trained mice (Materials and methods). (B) $\beta_{Muscimol}$ calculated for all 3 mice and 30 sets of sessions. Positive values represent leftward (ipsilateral) influence of muscimol. Arrowhead indicates $\beta_{Muscimol}$ for example sessions shown in A. (C) Behavioral influence of the previous choice calculated separately for muscimol and saline sessions (same sessions shown in B). *p=0.021, one-tailed Mann-Whitney *U* test.

(p=0.44 and p=0.28, respectively, two-tailed Mann-Whitney *U* tests), but that $\beta_0$ was higher in muscimol than saline sessions (p=0.040, two-tailed Mann-Whitney *U* test), indicating an odor-independent leftward bias consistent with *Figure 5B*]. These results support our hypothesis that the PPTg activity representing previous experience (*Figure 2*) influences action selection.

## Discussion

This study examined the neural basis by which action selection is influenced by recent experience. We first found that behavior in a stimulus-cued spatial-choice task can be accounted for by a Bayesian framework (among others) in which choices are influenced by both recent trial history and the sensory stimulus (*Figure 1*), consistent with previous studies (*Gold et al., 2008*; *Busse et al., 2011*; *Akaishi et al., 2014*). We then found that activity of a subpopulation of PPTg neurons reflects choices and outcomes in the recent past (*Figures 2* and *3*) and correlated with upcoming choice (*Figure 4*), suggesting a role in mediating the influence of recent experience on behavior. Furthermore, inactivating PPTg decreased the influence of recent experience on behavior (*Figure 5C*). Together, our results demonstrate that the PPTg encodes representations of recent experience that can contribute to action selection. Below, we consider our findings in light of the evolving view of the role of the PPTg within the interconnected network of brain regions involved in integrating priors with sensory evidence in order to select and control motor output (*Mena-Segovia et al., 2004*; *Gold and Shadlen, 2007*).

The motor-related role of the PPTg has traditionally been considered in the context of central pattern generation (*Garcia-Rill, 1991*). More recent studies in behaving animals have found that PPTg activity encodes specific actions and reflects their outcomes (*Matsumura et al., 1997*; *Dormont et al., 1998*; *Okada and Kobayashi, 2009*; *Norton et al., 2011*; *Maclaren et al., 2013*; *Thompson and Felsen, 2013*) and have suggested that the PPTg may be involved in attention and other cognitive processes (*Steckler et al., 1994*; *Winn, 2008*). The results described here are consistent with these findings and suggest that PPTg output may play a role in providing information about recent actions and their outcomes that can be used to guide the selection of actions mediated by downstream motor circuits. However, downstream circuits do not necessarily utilize this information, given that it is represented in PPTg activity even in mice that do not exhibit a behavioral dependence on previous choice (*Figure 1D*).

To where might PPTg representations of recent experience be conveyed in order to modulate action selection? While the PPTg projects to several motor-related regions including other brainstem nuclei and the striatum (*Beninato and Spencer, 1987*; *Inglis and Winn, 1995*), and the current study cannot rule out the involvement of these regions, an attractive candidate is the SC. Cholinergic PPTg neurons project to and excite neurons in the SC responsible for motor output (*Beninato and Spencer, 1986*; *Sooksawate et al., 2008*; *Stubblefield et al., 2015*), and nicotinic signaling in the SC has been shown to modulate orienting behavior (*Weldon et al., 1983*; *Aizawa et al., 1999*; *Watanabe et al., 2005*). The SC is thought to integrate a wide range of inputs in order to select orienting actions (*Kobayashi and Isa, 2002*; *Krauzlis et al., 2004*; *Grossberg et al., 2015*; *Wolf et al., 2015*) and is thus well-positioned to combine – possibly additively – incoming sensory information with prior representations of action value (provided before, during, or even after the onset of sensory input; *Figure 3B,C*) in order to select the most valuable action (*Trappenberg et al., 2001*; *Dorris et al., 2007*; *Kim and Basso, 2010*). Our results suggest that the source of these prior representations may be the PPTg, which could complement other SC inputs – such as inhibition from the basal ganglia (*Hikosaka and Wurtz, 1983*; *Chevalier et al., 1985*) – in biasing action selection by modulating SC processing according to recently-experienced action-value associations (*Kobayashi and Isa, 2002*; *Hikosaka et al., 2006*; *Wolf et al., 2015*). While we have found it useful

to think of the input from the PPTg to the SC as representing priors, in a Bayesian sense, accepting this framework – which may be imperfect given that choices in our task should not depend on previous trials – is not necessary for interpreting our results.

Given the importance of trial history for action selection, it is not surprising that representations of trial history similar to that shown here have been observed in several brain areas, including the striatum (*Lau and Glimcher, 2007*, *2008*; *Histed et al., 2009*; *Kim et al., 2013*), prefrontal cortex (*Histed et al., 2009*) and premotor cortex (*Marcos et al., 2013*). Activity in these regions typically reflects information from several previous trials, while PPTg activity was primarily associated with the immediately preceding trial (*Figure 2*). While this difference may be due to different task demands (recall that tracking past trials conferred no behavioral benefit in our task), the representations in the PPTg that we observed suggest a complementary mechanism to the more computationally expensive processing in these other regions for integrating internal representations of experience with sensory evidence: Specifically, that PPTg directly influences the circuits underlying action selection based on recent experience, which may be well-suited to the relevant dynamics of some real-world situations. Future studies can expand upon our findings by recording from specific types of neurons (e.g., cholinergic) in the PPTg (*Lima et al., 2009*; *Cohen et al., 2012*; *Roseberry et al., 2016*), as well as by examining how associations between stimuli and reward location are initially learned, in order to further elucidate the function of the PPTg in selecting actions.

## Materials and methods

### Animals

All experiments were performed according to protocols approved by the University of Colorado School of Medicine Institutional Animal Care and Use Committee. Mice were bred in the animal facilities of the University of Colorado Anschutz Medical Campus or purchased (Jackson Labs). We used 10 male adult mice, aged 124–186 days at the start of experiments. Pharmacology experiments were performed in three C57BL/6J mice, and electrophysiological experiments were performed in three C57BL/6J and four ChAT-Cre mice (Jackson Labs, strain B6; 129S6-*Chat*[tm2(cre)Lowl]) (we observed no differences across backgrounds and data are combined here). Mice were housed singly in a vivarium with a 12-hr light/dark cycle with lights on at 5:00 am. Food (Teklad Global Rodent Diet No. 2918; Harlan) was available ad libitum. Access to water was restricted to the behavioral session (see below) unless less than ~1 ml was received, in which case free water was provided for 2–5 min following the session (*Thompson and Felsen, 2013*).

### Behavior

Mice were trained on an odor-guided spatial choice task (*Uchida and Mainen, 2003*) as described in detail in *Thompson and Felsen (2013)*. Briefly, in each trial of the task, the mouse waited for a central port to be illuminated, entered the port, waited $144 \pm 64$ ms (mean $\pm$ s.d.) for the odor valve to open, sampled a binary odor mixture, waited $434 \pm 68$ ms (mean $\pm$ s.d.) for a go signal (simultaneous port light off and tone presentation; 5kHz, ~85dB), exited the odor port, and moved toward the left or right reward port [*Figure 1A*; these delays were selected in order to temporally segregate behavioral events for determining how they were correlated with neural activity, minimize training time, and for consistency with our previous work (*Thompson and Felsen, 2013*)]. In this previous work, we used a photo-ionization-detector to estimate the latency between odor valve open and the odor first arriving at the port to be 75–100 ms (*Thompson and Felsen, 2013*). Exiting the odor port prior to the go signal resulted in the unavailability of reward on that trial. All training and experimental behavioral sessions were conducted during the light cycle.

Odors were comprised of binary mixtures of (+)-carvone and (−)-carvone (Acros), commonly perceived as caraway and spearmint, respectively. In all sessions – including training on the task, as well as during neural recording and manipulation – mixtures in which (+)-carvone > (−)-carvone indicated reward availability at the left port, and (−)-carvone > (+)-carvone indicated reward availability at the right port. When (+)-carvone = (−)-carvone, the probability of reward at the left and right ports, independently, was 0.5. The full set of (+)-carvone/ (−)-carvone ratios used was 95/5, 80/20, 60/40, 50/50, 40/60, 20/80, 5/95. Mixtures were diluted in mineral oil and carrier air and delivered to the odor port at 800 ml/min. The mixture presented in each trial was selected pseudo-randomly. Reward

(5 µL water) was delivered by transiently opening a calibrated water valve 41 ± 32 ms (mean ± s.d.) after reward port entry. Mice completed training in 6–8 weeks and were then implanted with a drug-delivery cannula or a neural recording drive, as described below. Since our neural recording and manipulation experiments were performed in mice that were well-trained on the task, we do not attempt to examine how neural activity underlies task acquisition here (e.g., via reinforcement learning).

## Surgery

Two types of implants were used in these experiments: (1) a steel infusion cannula (Plastics One, Minneaspolis, MN) for muscimol delivery and (2) a Versa Drive 4 microdrive (containing four independently-adjustable tetrodes; Neuralynx) for tetrode recordings. All implants were targeted to the PPTg using the same general stereotactic procedure; implant-specific details are described below. Mice were placed in a ventilated chamber and briefly exposed to a volatile anesthesia (isoflurane, 2%; Priamal Healthcare Limited). Immediately following the onset of deep anesthesia (verified by toe-pinch), the mouse was placed in a stereotaxic device with a nose cone that continuously delivered 1%–1.5% isoflurane to maintain anesthesia. When the mouse was fully unresponsive to foot pinch and appeared to maintain a consistent breathing rate, the fur on the surface of the scalp was removed, and topical antiseptic (Betadine; Purdue Products) was applied along with ophthalmic ointment on the eyes. Before exposing the skull, a bolus of topical anesthetic (250 µL 2% lidocaine; Aspen Veterinary Resources) was injected under the surface of the scalp. With the skull exposed by central incision and scalp retraction, we adjusted head angle to align the elevation of bregma and lambda and drilled a 1.5 mm diameter cranial window centered on coordinates for the left PPTg (4.5 mm posterior from bregma, 1.1 mm lateral from the midline [*Paxinos and Watson, 2006*]). Following craniotomy and durotomy, we implanted one of the following:

### Infusion cannula

The steel guide and removable steel insert assembly were targeted to 0.15 mm dorsal to the surface of the PPTg. The cannula cap was affixed to the skull with two small screws (Plastics One), luting (3 M) and dental acrylic (A-M Systems).

### Versa Drive 4 microdrive

The drive was lowered into the craniotomy such that the tetrodes extending from its base targeted 100–200 µm above the dorsal surface of the PPTg (2.5 mm from brain surface). The drive was then affixed to the skull using two small screws, a luting, and dental acrylic. One screw was soldered to a ground wire connected to the drive.

In all surgeries, after the acrylic hardened, a topical antibiotic was applied to the scalp around the drive implant, the isoflurane was turned off and oxygen alone was delivered to the animal to gradually alleviate anesthesia. Immediately following surgery, animals were administered sterile 0.9% saline for rehydration and an analgesic (5 mg/kg Ketofen; Zoetis). Post-operative care, including analgesic and antibiotic administration, continued for up to 5 days after surgery and animals were closely monitored for signs of distress.

## Pharmacology

Prior to each session, an injection cannula was prepared with either muscimol (test sessions) or saline (control sessions) and inserted into the chronically implanted guide sans anesthesia. An infusion pump (Harvard Apparatus) was used to administer 150 nL of solution at 0.075 µL/min. Muscimol dosages ranged from 22 to 44 pmol. Mice recovered for at least 10 min before beginning the behavioral session.

## Electrophysiology

Recordings were collected using four tetrodes. Each tetrode consisted of four polyimide-coated nichrome wires (12.5 µm diameter; Sandvik) gold plated to 0.2–0.4 MΩ impedance. Electrical signals were amplified and recorded using the Digital Lynx S multichannel acquisition system in conjunction with Cheetah data acquisition software (Neuralynx). Tetrode depths, estimated by calculating the rotation of the screw affixed to the shuttle holding the tetrode (one rotation = ~250 µm), were

adjusted ~75 µm between recording sessions to sample independent populations of neurons across sessions. Offline spike sorting and cluster quality analysis was performed using MCLUST software (MClust-4.0, A.D. Redish et al.) in MATLAB. Briefly, single units were isolated by manual clustering based on features of the sampled waveforms (amplitude, energy, and the first principal component normalized by energy). Clusters with L-ratio <0.75 and isolation distance >12 were deemed single units (*Schmitzer-Torbert et al., 2005*), which resulted in excluding 30% of clusters. Although units were clustered blind to inter-spike interval (ISI), clusters with ISIs <1 ms were excluded.

## Histology

Final tetrode location was confirmed histologically using electrolytic lesions made after the last recording session and tetrode tracks (*Thompson and Felsen, 2013*). On day-one, post-lesion, mice were overdosed with an intraperitoneal injection of sodium pentobarbital (100 mg/kg; Sigma Life Science) and transcardially perfused with saline and ice-cold 4% paraformaldehyde (PFA) in 0.1 M phosphate buffer (PB). After perfusion, brains were submerged in 4% PFA in 0.1 M PB for 3.5 hr for post-fixation and then cryoprotected overnight by immersion in 20% sucrose in 0.1 M PB. The brain was embedded in optimal cutting temperature compound (ThermoFisher Scientific) and frozen rapidly on dry ice. Serial coronal sections (50 µm) were cut on a cryostat. Alexa 555 fluorescent Nissl (1:500, NeuroTrace; Invitrogen, catalog #N-21480) was used to identify cytoarchitectural features of the PPTg and verify tetrode tracks and lesions.

## Behavioral analysis

We quantified choice behavior with a logistic function of the form $p = \frac{1}{1+e^{-\eta}}$, where p is the choice made on a given trial (right choice, p=0; left choice, p=1; trials in which no reward port was entered within 1.5 s of odor port exit were excluded) and $\eta$ is the linear predictor, which was adapted to specific analyses as described below. For all analyses, $\eta$ consisted of at least $\eta_0 = \beta_0 + \beta_{Odor: L}x_{Odor: L} + \beta_{Odor: R}x_{Odor: R}$, where $\beta_0$ represents overall choice bias, $x_{Odor: L}$ and $x_{Odor: R}$ represent the strength of the odors associated with the left and right reward port, respectively, and $\beta_{Odor: L}$ and $\beta_{Odor: R}$ represent the influence of the odors on choice. $x_{Odor: L}$ and $x_{Odor: R}$ are calculated as (fraction of odor - 0.5) / 0.5 and range from 0 to 1; we used separate terms for left and right odors to allow for the possibility that they asymmetrically influenced choice. In *Figure 1E*, for the *Simple model*, $\eta = \eta_0$.

To assess the effect on choice behavior of inhibiting PPTg activity with muscimol (*Figure 5A,B*), we combined each muscimol session with its pre- and post-saline session and set $\eta = \eta_0 + \beta_{Muscimol}\,x_{Muscimol}$, where $x_{Muscimol} = 0$ for saline trials, $x_{Muscimol} = 1$ for muscimol trials, and $\beta_{Muscimol}$ represents the influence of muscimol on choice [positive values represent leftward (ipsilateral) influence] (*Salzman et al., 1992*). In addition, we examined the effect of inhibiting PPTg activity on movement latency and duration by examining the duration between the go signal and odor port exit, between odor port exit and reward port entry, and between reward port exit and odor port entry (to initiate the next trial).

To assess the effect on choice behavior of the choices and outcomes on previous trials, we set

$$\eta = \eta_0 + \sum_{n=1}^{N}\left(\beta^n_{Choice}x^n_{Choice} + \beta^n_{Outcome: L}x^n_{Outcome: L} + \beta^n_{Outcome: R}x^n_{Outcome: R}\right),\ \text{where}$$

$$x^n_{Choice} = \begin{cases} -1 & \text{for right choice} \\ 0 & \text{for no choice} \qquad n\,\text{trials back,} \\ 1 & \text{for left choice} \end{cases}$$

$$x^n_{Outcome: L} = \begin{cases} -1 & \text{for unrewarded left choice} \\ 0 & \text{for nonleft choice} \qquad n\,\text{trials back,} \\ 1 & \text{for rewarded left choice} \end{cases}$$

$$x^n_{Outcome: R} = \begin{cases} -1 & \text{for unrewarded right choice} \\ 0 & \text{for nonright choice} \qquad n\,\text{trials back,} \\ 1 & \text{for rewarded right choice} \end{cases}$$

$\beta^n_{Choice}$ represents the influence of the choice n trials back ('no choice' includes any excluded trials

and those in which odor port entry to initiate the next trial did not occur within 1.5 s of water port exit), $\beta^n_{Outcome:\,L}$ represents the influence of the outcome at the left $n$ trials back, and $\beta^n_{Outcome:\,R}$ represents the influence of the outcome at the right $n$ trials back. This set of coefficients provided the most intuitive interpretation of our data; our results did not depend on whether we instead included separate coefficients for previous left and right choice or a single coefficient for previous outcome. In *Figure 2D–F*, for the *Extended model, 1 trial back*, $n = 1$; for the *Extended model, 2 trials back*, $n = 2$.

To assess how inhibiting PPTg activity modulated the influence of the choice on the previous trial on behavior, we used a reduced form of the *Extended model, 1 trial back* by setting $\eta = \eta_0 + \beta^1_{Choice} x^1_{Choice}$, and calculated $\beta^1_{Choice}$ separately for inhibited (muscimol) and control (saline) sessions (*Figure 5C*).

## Preference analysis of neural data

To quantify the selectivity of single neurons for choice and outcome, we used an ROC-based algorithm (*Green and Swets, 1966*) that calculates the ability of an ideal observer to classify whether a given spike rate was recorded in one of two conditions (e.g., on trials in which the left or right reward port was selected). 'Preference' was calculated as $2(ROC_{area} - 0.5)$, a measure ranging from $-1$ to $1$, where $-1$ signifies the strongest possible preference for one alternative, $1$ signifies the strongest possible preference for the other alternative, and $0$ signifies no preference (*Thompson and Felsen, 2013*). For choice preference, $-1$ = left and $1$ = right; for outcome preference, $-1$ = no reward and $1$ = reward. Since choice is correlated with movement direction, this measure of choice preference alone formally cannot disambiguate whether activity reflects the chosen side itself or the direction of movement either towards or from the reward port. However, either movement is a perfect proxy for the choice – e.g., leftward movement towards the reward port and rightward movement from the reward port both indicate a left choice – and we report in the Results section an analysis derived from choice preference that can disambiguate whether PPTg activity reflects choice or movement direction. For clarity, we refer to neural activity with respect to the chosen reward port and the direction of movement towards that port (i.e., left and leftward in the above example). Statistical significance of preference was determined with a Monte Carlo permutation test: we recalculated preference after randomly reassigning each firing rate to one of the two groups, repeated this procedure 1000 times to obtain a distribution, and calculated the fraction of randomly generated preferences exceeding the actual preference. We examined preference during the pre-stimulus epoch (from odor port entry to 100 ms after odor valve open, before stimulus information can reach the PPTg; *Figure 1A*). For all preference analyses, we tested for significance at $\alpha = 0.05$. Neurons with fewer than four trials for each condition or with a firing rate below 2 spikes/s for both conditions were excluded from analysis (*Thompson and Felsen, 2013*).

## Regression analysis of neural data

To assess the influence on PPTg activity of the choices and outcomes on the previous and current trials (*Figure 3*), we fit the electrophysiological data with a multi-variable linear regression model of the form

$$FR(t) = \beta_0(t) + \beta^1_{Choice}(t)x^1_{Choice} + \beta^1_{Outcome}(t)x^1_{Outcome} + \beta^0_{Choice}(t)x^0_{Choice} + \beta^0_{Outcome}(t)x^0_{Outcome},$$

where $FR(t)$ is the mean firing rate in a given time bin $t$ (100 ms duration, sampled every 10 ms),

$$x^1_{Choice} = \begin{cases} -1 & \text{for an ipsilateral (left) choice} \\ 0 & \text{for no choice} \\ 1 & \text{for a contralateral (right) choice} \end{cases} \text{on the previous trial,}$$

$$x^1_{Outcome} = \begin{cases} -1 & \text{for a nonrewarded choice} \\ 0 & \text{for no choice} \\ 1 & \text{for a rewarded choice} \end{cases} \text{on the previous trial,}$$

$$x^0_{Choice} = \begin{cases} -1 & \text{for an ipsilateral choice} \\ 0 & \text{for no choice} \\ 1 & \text{for a contralateral choice} \end{cases} \text{on the current trial,}$$

$$x^0_{Outcome} = \begin{cases} -1 & \text{for a nonrewarded choice} \\ 0 & \text{for no choice} \\ 1 & \text{for a rewarded choice} \end{cases} \text{on the current trial,}$$

$\beta_0(t)$ represents the mean firing rate across trials in time bin $t$, and $\beta^1_{Choice}(t)$, $\beta^1_{Outcome}(t)$, $\beta^0_{Choice}(t)$, and $\beta^0_{Outcome}(t)$ represent the influence on firing rate of the previous choice, previous outcome, current choice and current outcome, respectively, in time bin $t$ (*Rorie et al., 2010*). Positive values for $\beta^1_{Choice}$ and $\beta^0_{Choice}$ indicate that firing rate is increased by contralateral choices and negative values indicate that firing rate is increased by ipsilateral choices. Positive values for $\beta^1_{Outcome}$ and $\beta^0_{Outcome}$ indicate that firing rate is increased by rewarded choices and negative values indicate that firing rate is increased by unrewarded choices.

## Acknowledgements

This work was supported by NINDS/NIH grant R01NS079518, the Boettcher Foundation's Webb-Waring Biomedical Research Award, and the University of Colorado Anschutz Medical Campus Optogenetics and Neural Engineering Core (P30NS048154). We thank Stephen V David, Brian Lau, and members of the Felsen lab for helpful discussions and comments on the manuscript.

## Additional information

### Funding

| Funder | Grant reference number | Author |
| --- | --- | --- |
| National Institute of Neurological Disorders and Stroke | R01NS079518 | Gidon Felsen |
| Boettcher Foundation | | Gidon Felsen |
| National Institute of Neurological Disorders and Stroke | P30NS048154 | Gidon Felsen |

The funders had no role in study design, data collection and interpretation, or the decision to submit the work for publication.

### Author contributions

JAT, JDC, Designed the experiments, Collected the data, Analyzed the data, Wrote the manuscript; GF, Designed the experiments, Analyzed the data, Wrote the manuscript

### Author ORCIDs

John A Thompson, [iD] http://orcid.org/0000-0003-2991-5194
Gidon Felsen, [iD] http://orcid.org/0000-0003-0745-8279

### Ethics

Animal experimentation: All experiments were performed according to protocols approved by the University of Colorado School of Medicine Institutional Animal Care and Use Committee (protocol #: B-90215(11)1D).

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
