## [Decision Letter]

Thank you for submitting your article "Mesencephalic representations of recent experience influence decision making" for consideration by *eLife*. Your article has been favorably evaluated by Sabine Kastner (Senior editor) and three reviewers, one of whom, Joshua Gold, is a member of our Board of Reviewing Editors.

The reviewers have discussed the reviews with one another and the Reviewing Editor has drafted this decision to help you prepare a revised submission.

Summary:

In this study, Thompson et al. trained mice to perform a task in which actions are guided by sensory inputs (odor), and are also influenced by actions and outcomes in the recent past (one trial in the past). They then examined the role of the pedunculopontine tegmental nucleus (PPTg) in this behavior. This group has previously shown that PPTg activity reflects the choice and outcome on the current trial. In the present study, they found that, in a time window before or around odor delivery, PPTg neurons represent the choice (left or right movement) or outcome (reward or no-reward) in the previous trial, but not in the current trial. They also pharmacologically or optogenetically manipulated the activity of PPTg, in an effort to establish a causal relationship between PPTg and the behavioral effect of the immediate recent choice/outcome.

The reviewers agreed that the questions were interesting, the approach was sound, the analyses were sophisticated and intuitive, the paper was clearly written, and many of the results were quite compelling. Figure 3 in particular was striking, showing a type of clear, one-trial-back tuning that is novel and will certainly be of interest to the field of decision making.

Essential revisions:

However, the reviewers also brought several major concerns. Primary among these was questions about the interpretability of the optogenetic results, given the modest effect size, apparent lack of certain controls, and other issues (see below for details). If and how those results should be presented in a revised manuscript should be considered carefully. We are concerned that doing the optogenetic work to the standard needed for publication in *eLife* will require significant new experiments, and it is against *eLife* policy to require substantial additional new experiments, although some authors take this on to make a paper as strong as it can be. So there are two options we see at this time: delete the optogenetic work or do the additional work needed to strengthen the optogenetics, and thus the paper. We encourage you to contact Josh Gold directly if you need further discussion or guidance on this point. Please note that if you choose to delete the optogenetic experiments, the other proposed revisions become crucial to enhance the depth and rigor of the data.

1) Concerning the optogenetic results:

A) Can the authors be sure that the optogenetic animals sufficiently and selectively expressed the channelrhodopsin? For example, in the subsection “Optical Stimulation” it states that the protocol was "shown in preliminary studies to be sufficient for driving neural activity." Is there a citation for this claim? If not published, it needs to be shown. Moreover, can they exclude the fact that heat from the optotrode had non-specific effects in the PPT? The authors state that they did post-mortem staining, but they don't show or report the results. Also, optotrode stimulation in a non-transgenic mouse, or data from others might be able to serve as a heat control.

B) The optogenetic experiment also should be paired with a control experiment in which mice with GFP expressed in PPTg ChAT+ cells go through exactly the same procedures as the experimental groups.

C) It is not clear why the stimulation was performed on ChAT+ cells; the authors need to first show that these ChAT+ cells have relevant response properties, i.e., that they show modulation by choice/outcome in the previous trial.

D) The result that there was no difference between light-on and light-off trials when stimulating the ChAT+ cells is against the hypothesis that these neurons are important for mediating the behavioral effect of previous choice/outcome. Instead, light-stimulation affected all trials in the entire session. This result suggests that light-stimulation caused a non-specific effect; e.g. it might have globally changed the behavioral state of the animal, such as their sensitivity to outcomes, attentional state, etc., during the stimulated session. This is particularly a problem when a large number of trails are stimulated, which is the case of this study (40% of trails were stimulated).

E) There was no difference between light-stimulation during the pre-stimulus/odor epoch and that during the epoch after odor valve open. This result also does not support the major hypothesis that these neurons are important for mediating the behavioral effect of previous choice/outcome.

F) In order to establish a causal relationship between PPTg neuronal activity and the behavioral effect of the immediate recent choice/outcome, the authors need to do optogenetic inhibition specifically before odor onset. The pharmacological inhibition does not have the temporal resolution to selectively target the activity representing previous choice.

G) Given the modest optogenetic results, it might be useful to present a bit more information to make it easier to assess the reliability of those results. In particular, would it be possible to present the results in a way that shows the per-animal results? In other words, were the differences driven by just one animal?

2) The muscimol experiment in Figure 1 seems to be disconnected from the rest of the paper. It was not clear until the end of the paper that the authors tried to make a conclusion that this manipulation affected the effect of previous choice. However, as discussed above, this conclusion is based largely on the optogenetic results, which are not convincing.

3) Figure 4: it would be helpful to include a panel showing the population data as not just fractions of significant units (as in panels B and C), but also as the mean{ ± }SEM coefficient estimates (as in Figure 4), to get a better sense of the magnitude of the effects across the population.

4) A number of important experimental and analysis details should be included. For example, were neural data from animals M1 and M2 included, despite the fact that neither had behavior that was influenced by the prior trial? This should be explained and justified better. Moreover, the authors should put in the main text: how many trials were run per session, how many sessions were run per animal, and per condition, how many animals were included in every given analysis, and the total number of single units analyzed. The authors also state that they excluded sessions with the tip fibers outside the PPT – how many were these? Was one of the three optogenetic animals entirely excluded (or did these sessions serve as control stimulation outside the PPT?)?

---

## [Author Response]

*Essential revisions:*

*1) However, the reviewers also brought several major concerns. Primary among these was questions about the interpretability of the optogenetic results, given the modest effect size, apparent lack of certain controls, and other issues. If and how those results should be presented in a revised manuscript should be considered carefully. We are concerned that doing the optogenetic work to the standard needed for publication in eLife will require significant new experiments, and it is against eLife policy to require substantial additional new experiments, although some authors take this on to make a paper as strong as it can be. So there are two options we see at this time: delete the optogenetic work or do the additional work needed to strengthen the optogenetics, and thus the paper. We encourage you to contact Josh Gold directly if you need further discussion or guidance on this point. Please note that if you choose to delete the optogenetic experiments, the other proposed revisions become crucial to enhance the depth and rigor of the data.*

We thank the Editors for their evaluation of the optogenetics experiments and for suggesting these two options. We recognize that much more work could be done to strengthen the optogenetic results such that they would be suitable for publication as part of this manuscript. However, we agree that doing so would require substantial new experiments. While these are worthwhile experiments and we are confident that they would yield interesting data, performing the necessary experimental work is not feasible at this time. We have therefore decided to delete the optogenetic experiments from this manuscript. Thus, we do not address the optogenetics-related comments here, but we of course address all other concerns and have made every effort to enhance the depth and rigor of the data. Throughout the revised manuscript, we have removed text related to the optogenetics experiments from the Abstract, Introduction, Methods, Results, Discussion and Figures.

*2) The muscimol experiment in Figure 1 seems to be disconnected from the rest of the paper. It was not clear until the end of the paper that the authors tried to make a conclusion that this manipulation affected the effect of previous choice. However, as discussed above, this conclusion is based largely on the optogenetic results, which are not convincing.*

The intent of our muscimol experiments was to show that unilaterally inactivating PPTg affects behavior, demonstrating a causal relationship between PPTg activity and behavior. In the revised manuscript, we have reorganized our presentation of the muscimol experiments such that they are all now at the end of the Results (Revised Figure 5; Results, last two paragraphs) instead of being split between Original Figure 1 (direct effect of inactivation on behavioral choice; now Revised Figure 5) and Original Figure 6D (effect of inactivation on the influence on behavior of the previous choice; now Revised Figure 5). Now that we have deleted the optogenetics experiments, we believe that this organization provides the most natural flow for the manuscript: Revised Figure 2–Figure 4 demonstrate correlation, and Revised Figure 5 demonstrates causality, between PPTg activity and behavior.

*3) Figure 4: it would be helpful to include a panel showing the population data as not just fractions of significant units (as in panels B and C), but also as the mean{ ± }SEM coefficient estimates (as in Figure 4), to get a better sense of the magnitude of the effects across the population.*

In Revised Figure 3 (adapted from Original Figure 4), we now include two new panels that show mean ± SEM coefficient estimates across the population for choice (Revised Figure 3) and outcome (Revised Figure 3). Interestingly, displaying the data this way reveals that the preferred previous trial choice of many neurons briefly inverted immediately preceding movement initiation (Revised Figure 3), consistent with the dip in the fraction of significant neurons at this time (Revised Figure 3). We note this observation in the revised manuscript (subsection “Influence of previous trials on PPTg activity”, last paragraph).

*4) A number of important experimental and analysis details should be included. For example, were neural data from animals M1 and M2 included, despite the fact that neither had behavior that was influenced by the prior trial? This should be explained and justified better.*

We clarify in the revised manuscript (subsection “Influence of previous trials on PPTg activity”, first paragraph) that neural data (shown in Revised Figure 2, Figure 3 and Figure 4) were included from all mice for which behavioral data were included (shown in Revised Figure 1). Thus, data from M1 and M2 were included, even though these mice did not exhibit behavior influenced by the previous trial (Revised Figure 1). Indeed, we took advantage of the absence of a behavioral effect in these mice to show that the relationship between firing rate and current choice (Revised Figure 4) was not indirectly due to the fact that both of these variables depend on previous choice (Firing rate: Revised Figure 2; Current choice: Revised Figure 1): Even for M1 and M2, in which current choice did not depend on previous choice, more neurons exhibited a positive than negative relationship between firing rate and current choice (Revised Figure 4; subsection “Influence of PPTg representations of previous trials on behavior”, first paragraph). In the revised manuscript, we have edited our conclusion from this analysis to “These results suggest that the representation of the choice on the previous trial (Figure 2) can directly influence the choice made by the mouse on the current trial”, and have likewise edited text in the Discussion to “Together, our results demonstrate that the PPTg encodes representations of recent experience that can contribute to action selection”.

Finally, in the revised manuscript we speculate that these mice exhibited a neural, but not a behavioral, effect because the PPTg provides information about the previous trial to downstream regions (e.g., the superior colliculus), but that “downstream circuits do not necessarily utilize this information, given that it is represented in PPTg activity even in the absence of a behavioral dependence on previous choice (Figure 1)” (Discussion).

*Moreover, the authors should put in the main text: how many trials were run per session, how many sessions were run per animal, and per condition, how many animals were included in every given analysis, and the total number of single units analyzed.*

In the revised manuscript we have included substantial new experimental details: a) For the behavioral and recording experiments (Revised Figure 1–Figure 4), each of the 7 mice performed 12 sessions (subsection “Influence of previous trials on behavior”, second paragraph). b) In these sessions, mice performed 451 ± 115 trials per session (mean ± STD; in the aforementioned paragraph). c) For the muscimol experiments (Revised Figure 5), each of the 3 mice performed 10 sessions per condition [saline (pre), muscimol, and saline (post)], for 30 total sessions per mouse (subsection “Influence of PPTg representations of previous trials on behavior”, second paragraph). d) In these sessions, mice performed 424 ± 105 trials per session (mean ± STD; in the aforementioned paragraph). e) We recorded from 506 well-isolated PPTg neurons (subsection “Influence of previous trials on PPTg activity”, first paragraph), data from all of which are included in Revised Figure 2. Revised Figure 3 and Figure 4 include data from the 378 neurons that exhibited a significant preference for choice on the previous trial (corresponding to the black bars in Revised Figure 2; subsection “Influence of previous trials on PPTg activity”, last paragraph and subsection “Influence of PPTg representations of previous trials on behavior”, first paragraph.